# Chrysophanol-Induced Autophagy Disrupts Apoptosis via the PI3K/Akt/mTOR Pathway in Oral Squamous Cell Carcinoma Cells

**DOI:** 10.3390/medicina59010042

**Published:** 2022-12-26

**Authors:** Dan-Bi Park, Bong-Soo Park, Hae-Mi Kang, Jung-Han Kim, In-Ryoung Kim

**Affiliations:** 1Postech-Catholic Biomedical Engineering Institute, College of Medicine, The Catholic University of Korea, Seoul 06591, Republic of Korea; 2Department of Oral Anatomy, School of Dentistry, Pusan National University, Yangsan 50612, Republic of Korea; 3Dental and Life Science Institute, School of Dentistry, Pusan National University, Yangsan 50612, Republic of Korea; 4Department of Oral and Maxillofacial Surgery, Medical Center, Dong-A University, Busan 49201, Republic of Korea

**Keywords:** apoptosis, autophagy, chrysophanol, oral squamous cell carcinoma cells

## Abstract

*Background and Objectives*: Natural products are necessary sources for drug discovery and have contributed to cancer chemotherapy over the past few decades. Furthermore, substances derived from plants have fewer side effects. Chrysophanol is an anthraquinone derivative that is isolated from rhubarb. Although the anticancer effect of chrysophanol on several cancer cells has been reported, studies on the antitumor effect of chrysophanol on oral squamous-cell carcinoma (OSCC) cells have yet to be elucidated. Therefore, in this study, we investigated the anticancer effect of chrysophanol on OSCC cells (CAL-27 and Ca9-22) via apoptosis and autophagy, among the cell death pathways. *Results*: It was found that chrysophanol inhibited the growth and viability of CAL-27 and Ca9-22 and induced apoptosis through the intrinsic pathway. It was also found that chrysophanol activates autophagy-related factors (ATG5, beclin-1, and P62/SQSTM1) and LC3B conversion. That is, chrysophanol activated both apoptosis and autophagy. Here, we focused on the roles of chrysophanol-induced apoptosis and the autophagy pathway. When the autophagy inhibitor 3-MA and PI3K/Akt inhibitor were used to inhibit the autophagy induced by chrysophanol, it was confirmed that the rate of apoptosis significantly increased. Therefore, we confirmed that chrysophanol induces apoptosis and autophagy at the same time, and the induced autophagy plays a role in interfering with apoptosis processes. *Conclusions*: Therefore, the potential of chrysophanol as an excellent anticancer agent in OSCC was confirmed via this study. Furthermore, the combined treatment of drugs that can inhibit chrysophanol-induced autophagy is expected to have a tremendous synergistic effect in overcoming oral cancer.

## 1. Introduction

Oral cancer comprises malignant neoplasms that accrue from the tongue, lip, alveolar, oral cavity, oropharynx, and nasopharynx [1]. Globally, the number of newly diagnosed oral cancer patients is around 360,000, and deaths numbered 180,000 in 2018 [2]. Oral cancer is commonly defined as an OSCC, and more than 90% are squamous-cell carcinomas present in the mucous membranes [3]. Chemotherapy, radiation therapy, and surgery are the generally used methods for cancer management. Surgery is primarily performed for oral cancer treatment, but in an event of recurrence of cancer after surgery, the prognosis is poor [4]. Postoperative adjuvant chemotherapy might be better for preventing relapses in patients than performing surgeries alone with respect to esophageal cancer [5]. Similarly, chemotherapy is required for the treatment of oral cancer to prevent recurrences. Unfortunately, there is no particularly effective chemotherapy for oral cancer in recent years [6]. Therefore, the development of chemotherapy is necessary for the prevention of recurrence and the advances of operation methods.

Apoptosis plays a critical role in regulating cell death in normal development and homeostasis [7]. Appropriate apoptotic signaling is important for maintaining the equilibrium between cell survival and death [8]. In many cancers, apoptosis evasion is an important reason for drug resistance [9]. Autophagy is a catabolic process induced by various cellular stresses, such as starvation and organelle damage [10]. Autophagy is featured by sequestrating the cytoplasmic contents and organelles within double- or multi-membrane autophagic vesicles called autophagosomes [11]. An autophagosome fused with a lysosome then degrades engulfed cytoplasm contents, such as damaged organelles, aggregated molecules and misfolded proteins; and recycles some autophagic cargo [12]. As a stress response, autophagy accompanies most processes for survival, such as maintaining homeostasis, rather than promoting cell death, which rarely occurs during a failed survival attempt [10]. However, clearly determining whether autophagy is a tumor promoter or tumor suppressor is difficult [13]. Previous studies suggest that autophagy plays a contrary role in cancer development. Autophagy is activated as a tumor suppressor in the early phase of cancer formation and seems to activate similarly to a tumor promoter if the tumor’s progression is significantly advanced [14]. Various mechanisms are involved in apoptosis and autophagy. In most cases, apoptosis is a “self-killing” signal that leads to cell death, and autophagy is a “self-eating” signal that absorbs unwanted proteins or organelles as nutrients to prolong cell survival. In other words, autophagy plays a role in protecting cells by preventing the process of cells proceeding toward apoptosis. On the other hand, in some cases, autophagy develops as a primary response to stress stimuli, followed by apoptosis or necrotic cell death [15].

The phosphatidylinositol-3 kinases (PI3Ks)/protein kinase B (Akt) signaling pathway serves as a pivotal key regulator of cell survival, proliferation, apoptosis, and autophagy in cancer cells [16]. PI3K is composed of four classes: PI3K-IA, PI3K-IB, PI3K-II, and PI3K-III. The activated PI3K helps the inositol ring 3′-OH group’s phosphorylation in inositol phospholipids and converts phosphatidylinositol (4,5)-bisphosphate (PIP2) to phosphatidylinositol (3,4,5)-bisphosphate (PIP3), and then recruits downstream. Akt is recognized as a major mediator of the downstream of PI3K [17,18]. Akt activations play anti-apoptotic roles, inhibiting Bcl-2 pro-apoptotic family members (including Bad, Bax, and Bim); on the other hand, Bcl-2 and Bcl-XL belong to the anti-apoptotic family, which is promoted via the activation of NF-κB transcription [19]. Moreover, the activated Akt promotes the phosphorylated and activates mammalian target of rapamycin (mTOR) [20]. The mTOR’s activation performs pleiotropic roles in the function of cell death, mediating apoptosis and autophagy. mTOR controls apoptosis-regulatory proteins such as p53, Bad, and Bcl-2 to promote cell death; meanwhile, the upregulation of autophagy, such as the ULK complex’s phosphorylation, prolongs cell survival [21,22]. Moreover, the Akt/mTOR pathway significantly regulates various hallmarks of cancer, such as survival, proliferation, apoptosis, autophagy, metastasis, and angiogenesis in OSCC cells [23]. Akt/mTOR inhibitors could come into the spotlight as therapeutic targets in OSCC.

Natural products provided a necessary source for the discovery of anti-cancer chemotherapy over the past few decades [24,25,26,27,28,29]. Chrysophanol (1,8-dihydroxy-3-anthraquinone) (Figure 1) is derived from *Rheum rhabarbarum* (rhubarb) and is used in traditional medicine in Asia [30]. Over the years, the pharmacological properties of chrysophanol have been researched; for example, resistant effects relative to inflammatory diseases, diabetes, bacterial, oxidation, ulcer, and cancer have been confirmed [31,32]. In addition, chrysophanol has shown potent anti-cancer effects on some cancers, such as liver, colon, and ovarian cancers [33,34,35,36,37,38,39].

Several studies have found that chrysophanol increases apoptosis in various cancer cells, but the mechanisms of apoptosis and autophagy induced by chrysophanol in oral cancer cells and their interrelationships have not yet been studied. In this study, we investigated the effects of chrysophanol on apoptosis and autophagy mechanisms, and furthermore, whether autophagy plays a role in protecting cells in the process of chrysophanol-induced apoptosis or autophagy as the primary response of apoptosis relative to stress stimuli induced by chrysophanol.

## 2. Materials and Methods

### 2.1. Regents

Chrysophanol was obtained from Sigma Aldrich (St. Louis, MO, USA). It was diluted with dimethyl sulfoxide (DMSO) and kept frozen. The stock was diluted to different concentrations with media as needed for the experiments. Crystal violet, 3-methyladenine (3-MA), LY294002, monodensylcadaverine (MDC), and acridine orange (AO) were purchased from Sigma Aldrich (St. Louis, MO, USA). JC-1 iodide (JC-1) was obtained from Santa Cruz (CA, USA). Specific antibodies for mTOR, p-mTOR, ATG5, beclin-1, cytochrome c, PARP, and caspase 3 were from Cell Signaling Technology (Danvers, MA, USA). LC3B and p62 were purchased from Sigma Aldrich (St. Louis, MO, USA); and Akt, p-Akt, Bax, Bcl-2, and β-actin were purchased in Santa Cruz (CA, USA). Secondary antibodies of mouse anti-rabbit IgG and rabbit anti-mouse IgG antibodies were obtained from Enzo Biochem (Farmingdale, NY, USA). The TOPscrip^tTM^ cDNA synthesis kit and TOPreal^TM^ qPCR 2× PreMIX (SYBR Green with Low ROX) were procured from Enzynomics (Dajeon, Republic of Korea). PCR primers (ATG5, Beclin-1, LC3B, and GAPDH) were obtained from macrogen (Seoul, Republic of Korea).

### 2.2. Cell Culture

CAL-27 and Ca9-22 cells are frequently used cell lines in the field of human oral squamous-cell carcinoma (OSCC) and were obtained from ATCC (Rockveile, MD, USA). CAL-27 and Ca9-22 were cultivated in Dulbecco’s modified eagle medium (DMEM) with 10% fetal bovine serum (FBS) (GE-Healthcare, Chicago, IL, USA) and 1% penicillin–streptomycin at 37 °C in 5% humidified CO_2_. The cells were grown in culture media at 70% confluence in the culture dishes.

### 2.3. Cell Proliferation Assay

The MTT assay was conducted to determine the cytotoxicity effects of chrysophanol. In a 96-well plate, both cell lines were grown at 1 × 10⁴ cells/well and treated with various doses of chrysophanol (0–300 μM) for 24 and 48 h. After the treatment with chrysophanol, the media were removed, and 0.05 mg/mL of an MTT solution was added to the media and incubated until the formation of formazan crystals at 37 °C. The formed formazan crystals were liquefied with DMSO, and absorbance was measured at 570 nm using the SpectraMax iD3 (BioTek, Winooski, VT, USA) and calculated as a percentage.

### 2.4. Colony-Formation Assay

On each well of a 6-well plate, cells were seeded and treated with different doses of chrysophanol for seven days. Methanol at 100% was used to fix cell colonies, and they were dyed with a 1% crystal violet solution for 10 min. Next, the colonies were washed with distilled water three times and dried. The number of colonies was counted using an optical microscope and calculated as a percentage.

### 2.5. Fluorescence Images

To obtain fluorescence images using Hoechst 33342, JC-1, MDC, and AO, cells (1 × 10⁴) were seeded in a Greiner bio-one 96-well plate (Greiner, Kremsmünster, Austria) and cultured for 24 h until the cells adhered to the well and stabilized. Then, chrysophanol was added to the medium according to the determined concentration, applied to the cells, and cultured for 24 h. Hoechst 33342 was used to observe nuclear morphological changes. Cells were fixed with 4% paraformaldehyde 10 min, and 1 μg/mL of a Hoechst 33342 solution stained the nuclei of the cells at 37 °C. JC-1 was used for verifying the change of mitochondria membrane potential (ΔΨm). The JC-1 dye (2 μg/mL) stained for 30 min at 37 °C. To identify acidic vacuoles, an acridine orange (AO) agent (1 μg/mL) was used to stain the acidic vesicular organelles for 5 min. For autophagic fluorescence observation, the final concentration of 50 μM of the monodansylcadaverine (MDC) solution was stained with autophagic vacuoles for 30 min. After staining, cells were washed 3 times with PBS and mounted using glycerol. Lionheart FX Automated Microscope (BioTek, Winooski, VT, USA) was used to observe changes in cellular fluorescence images. Cells were observed and photographed at ×200 magnification.

### 2.6. Gene Expression Analysis by Real-Time PCR

Cells were seeded on a six-well plate at a density of 2 × 10^5^ cells per well and incubated for 24 h to allow the cells to adhere and stabilize in the culture dish. Then, chrysophanol at a defined concentration was applied to the cells, cultured for 24 h, and then harvested. The total RNA was prepared using the RNeasy Mini Kit (Qiagen Inc., Valencia, CA, USA). The total RNA concentration was determined with a microplate spectrophotometer using the SpectraMax iD3 micro reader (BioTek, Winousk, VT, USA). cDNA was synthesized using a total RNA (2 μg) with a TOPscript^TM^ cDNA synthesis kit (Enzynomics, Dajeeon, Republic of Korea) by following the manual. Subsequently, quantitative real-time PCR was executed using the TOPreal^TM^ qPCR 2× PreMIX (SYBR Green with Low ROX) (Enzynomics, Dajeeon, Republic of Korea) with QuantStudio 1 (Applied Biosystems, Foster City, CA, USA). The relative mRNA levels were normalized using GAPDH as a housekeeping gene. The PCR primer sequences are shown in Table 1.

### 2.7. Western Blot Analysis

Cells (1 × 10^6^) were seeded on a 100 mm culture dish and incubated for 24 h to allow the cells to adhere and stabilize in the culture dish. Then, chrysophanol at a defined concentration was applied to the cells, cultured for 24 h, and then harvested. Cells were lysed in a RIPA (radioimmunoprecipitation assay) buffer (Cell signaling, Danvers, MA, USA), which was mixed with 2 mM PMSF and a 10 μL/mL protein inhibitor cocktail for 2 h. The samples were centrifuged at 13,200 rpm for 30 min. The Bradford protein assay was used to quantify the collected proteins, and 20 μg of protein was used to make loading samples. Samples were loaded based on molecular weights by gel-electrophoresis (SDS-PAGE) with 10% and 12.5% gels at 80 V for 2 h, and then the loaded gels were transferred onto a polyvinylidene difluoride (PVDF) membrane at 20 V for 16 h. The PVDF membrane was incubated with the appropriate primary antibodies blended at 1:1000 in a 5% non-fat dry milk solution overnight at 4 °C. Sequentially, the membrane was washed 5 times for 1 h with a Tris-NaCl-EDT (TNE) buffer and incubated with secondary antibodies at room temperature. Next, the membrane was washed 5 times for 1 h with a TNE buffer again, and the detection of protein was performed using a super signal West Femto (Pierce, Illinois, USA). The protein expression was detected with ImageQuant LAS 500 chemiluminescence (GE Healthcare, Chicago, IL, USA).

### 2.8. Immunofluorescence

The cells (2 × 10^4^) were placed in an 8-well Lab-Tek II Chambered Slide (Invitrogen, Thermo Fisher Scientific, Waltham, MA, USA) and incubated for 24 h to allow the cells to adhere and stabilize in the chambered slide. After chrysophanol treatments, the cells were washed with PBS and stained with 100 nM Mitotracker Deep Red at 37 °C for 1 h. Then, cells were fixed with 4% paraformaldehyde for 30 min and permeabilized with 0.1% Triton X-100 solved in PBS for 10 min. The cells were blocked with 1% BSA-PBS for 1 h and incubated with cytochrome c antibody (1:100) in 1% BSA-PBS with 0.1% Tween 20 at 37 °C for 2 h. Next, the cells were washed with PBS five times for 10 min and then incubated with secondary antibodies conjugated to Alexa Fluor (Alexa 488) (1:100) in 1% BSA-PBS for 2 h. Nuclear staining used the Prolong^TM^ Gold antifade reagent with DAPI (Invitrogen, Carlsbad, CA, USA). Finally, fluorescent images were observed and analyzed using a Zeiss LSM 700 laser-scanning confocal microscope (Cal Zeiss, Göettingen, Germany).

### 2.9. Statistical Analysis

GraphPad Prism version 5.0 (San Diego, CA, USA) was used for the statistical analysis. The one-way and two-way ANOVA were used to calculate significance. Significance was assumed to be reached at *** *p <* 0.001, ** *p <* 0.01, and * *p <* 0.05. The graph bar express mean ± SD.

## 3. Results

### 3.1. Chrysophanol Reduced Cell Viability and Proliferation in OSCC Cells

CAL-27 and Ca9-22 cells were treated with various concentrations of chrysophanol (0–300 μM) for 24 and 48 h, and then an MTT assay was conducted for both cells to confirm the cytotoxic effect of chrysophanol. After treatment with chrysophanol, the cell’s viability was reduced in a dose-dependent manner (Figure 2A,B). The IC50 values for each hour in Cal 27 cells were 230.6 µM (24 h), 177.6 µM (48 h), and 152.1 µM (72 h) of chrysophanol. In Ca9-22 cells were 227.1 µM (24 h),169.3 µM (48 h), and 154.4 µM (72 h) of chrysophanol. Cell proliferation was assessed by a colony-formation assay. Both cells were incubated with a low dose of chrysophanol (0–150 μM) for 7 days, and then the number of colonies was counted and calculated as a percentage in the histogram. It was confirmed that the number of colonies was reduced in chrysophanol-treated cells. Chrysophanol treatments dose-dependently inhibited cell proliferation in both types of cells (Figure 2C–F). The results clearly demonstrated that chrysophanol reduces cell viability and proliferation in OSCC cells.

### 3.2. Chrysophanol-Induced Apoptosis via the Caspase Activation in OSCC Cells

During apoptosis, cell morphology was altered by cell shrinkage, chromatin condensation, and apoptotic bodies [40]. Hoechst staining was conducted to observe the morphology changes due to chrysophanol in CAL-27 and Ca9-22 cells. Chrysophanol-treated cells were confirmed to have blue fluorescence, which indicates fragmented and condensed nuclei. Thus, the chrysophanol-treated cells had more morphological changes compared to the control (Figure 3A,B). The JC-1 dye was used as a practical tool to measure the mitochondrial membrane potential (∆ψM). Green fluorescence is an indicator of monomers, and red fluorescence is an indicator of J-aggregates [41]. As a result of JC-1 staining, chrysophanol-treated cells had higher expression in terms of red fluorescence compared to the control (Figure 3C,D). To analyze the signaling molecules closely related to apoptosis, Western blot analysis was performed for CAL-27 and Ca9-22 cells. Both cell types were treated with chrysophanol and incubated for 24 h. Protein expression levels that can indicate apoptosis, such as Bax, Bcl-2, caspase-3, caspace-7, and poly (ADP-ribose) polymerase (PARP), were then measured by a Western blot analysis in both cell types. Chrysophanol-treated cells showed caspase-3 and -7 activation, cleaved caspase 3, and PARP upregulation; and anti-apoptotic proteins and Bcl-2 were downregulated (Figure 3E). The relative protein ratios of PARP/cleaved PARP and Bax/Bcl-2 increased in a dose-dependent manner (Figure 3F,G). These results indicate that chrysophanol treatments significantly increase apoptosis via the caspase activation in OSCC cells.

### 3.3. Chrysophanol-Induced Autophagy in OSCC Cells

Acidic vesicular organelle (AVO) formations have autophagic characteristics [42]. For the detection of AVOs, vital staining with acridine orange (AO) was examined in CAL-27 and Ca9-22 cells by fluorescence microscopy. The control cells presented mostly green fluorescence, indicating the absence of AVOs. Chrysophanol-treated cells had more red fluorescence and minimal green fluorescence. Mature autophagic vacuoles, such as autolysosomes, stained with monodansylcadaverine (MDC), were observed as distinct blue dots within the cytoplasm or perinuclear regions [11]. To detect the chrysophanol effects on the formation of mature autophagic vacuoles, cells were stained with MDC and then observed using a fluorescence microscope. In chrysophanol-treated cells, autophagic vacuole formation increased in quantity and size compared with the control (Figure 4A,B). Next, the expression levels of proteins that indicate autophagy, such as ATG5, beclin-1, and LC3B, were measured by a Western blot analysis. Cells were treated with chrysophanol for 24 h. The expression levels of beclin-1 and ATG5 were upregulated and converted LC3B-I to LC3B-II, which increased dose-dependently in CAL-27 and Ca9-22 cells (Figure 4C,D). The expression level of mRNA (ATG5, p62/SQSTMI, and MAP1LC3B) was examined using real-time PCR. ATG5, p62/SQSTMI, and MAP1LC3B mRNA levels were dose-dependently upregulated in both cell types. These results revealed that autophagy-related molecules are regulated by chrysophanol in both cells (Figure 4E–G). Therefore, these experiments provided evidence that chrysophanol treatments cause autophagy in OSCC cells.

### 3.4. Chrysophanol-Induced Autophagy Impeded Apoptosis in OSCC Cells

3-Methyladenine (3-MA), an autophagy inhibitor [43], was used to investigate the effects of autophagy and apoptosis in CAL-27 and Ca9-22 cells. Both cells were treated with 2 mM 3-MA in advance and chrysophanol in the presence or absence of 3-MA. Cells with a single treatment of chrysophanol reflected a higher rate of cell viability than a combination of chrysophanol and 3-MA (Figure 5A,B). Western blot analysis was performed to determine the protein expression levels. The autophagy-related protein beclin-1 expression and the LC3B conversion level were reduced in the 3-MA combination group (Figure 5C). Chrysophanol treatments with 3-MA induced the expression of apoptosis-related protein caspase-3 activation and cleaved PARP, and the ratios of cleaved caspase3/caspase 3 and cleaved PARP/PARP improved dose dependently (Figure 5D–G). When the cells were undergoing apoptosis, the release of cytochrome c activated the caspase cascade [44]. Cytochrome *c* was shown by immunofluorescence, and it was observed more in co-treatments than in a single treatment. Inhibited autophagy induced the eruption of cytochrome *c.* (Figure 5H,I). The autophagic process was mediated by anti-apoptotic signals. The results showed that chrysophanol-induced autophagy interrupted apoptosis in OSCC cells.

### 3.5. Chrysophanol Produced the Akt/mTOR Signaling Pathway in OSCC Cells

Chrysophanol dose-dependently prompted the phosphorylation of Akt in both cells. The protein expression levels of p-Akt and p-mTOR dose-dependently increased in both cells (Figure 6A,B). The PI3K inhibitor, LY294002, was used as a tool to measure PI3K/Akt/mTOR signaling pathways. The cells were pre-treated in the presence or absence of 20 μM of LY294002 for 2 h and then treated with 150 μM of chrysophanol for 24 h. LY294002 treatment cells reflected the downregulation of p-Akt and p-mTOR protein expression (Figure 7A–C). LY294002 reduced the protein expression levels of beclin-1 and decreased the conversion of LC3B-I to LC3B-II. In contrast, inhibiting the PI3K/Akt pathway induced a caspase cascade, improving Bax and activating caspase-3 and PARP while diminishing Bcl-2. Thus, the cells that were treated with LY294002, and chrysophanol expressed more proteins that were associated with activating apoptosis and autophagy than a single treatment of chrysophanol (Figure 7D–G). As a result of staining with the JC-1 dye, the cells that were treated with LY294002 and chrysophanol had a lower ∆ψM than single-treatment cells (Figure 7H,I).

## 4. Discussion

Natural chemicals extracted from plants are considered good therapeutic agents for cancer therapy because they have fewer side effects and can be used for oral administration [45,46]. Previous studies indicated that chrysophanol induces cell death via the production of ROS and damaged ATP synthesis in liver and ovarian cancer [37,38,47]. Furthermore, chrysophanol has shown anti-cancer effects by anti-proliferation and pro-apoptotic activities in several cancers, such as choriocarcinoma, breast, and colon cancers [34,36,39]. However, studies on the autophagy effect of chrysophanol on cancer cells are still insignificant. In this study, as a first step toward demonstrating whether chrysophanol affects apoptosis and autophagy activities, our researchers identified an interaction between chrysophanol-induced apoptosis and autophagy in OSCC cells.

The results indicated that chrysophanol reduced cell viability and proliferation in CAL-27 and Ca9-22 cells (Figure 2). Chrysophanol inhibits cell proliferation by the inhibition of the NF-κB and EGFR/mTOR pathways in colon and breast cancers [34,39]. It has been confirmed that the treatment of chrysophanol caused apoptosis via increased morphological changes in OSCC cells. The mitochondrial signaling pathway of apoptosis plays a key role in regulating cell death in response to various stimuli [48]. During apoptosis, mitochondrial outer membrane permeability metastasis is induced by a specific Bcl-2 family [49]. We detected that chrysophanol induced changes in the mitochondrial membrane potential (∆ψM) and indicated that chrysophanol decreased ∆ψM.

If ∆ψM is lower, it could be an apoptotic indicator of the mean of emission apoptogenic factors [50]. Inhibited Bcl-2 protein releases cytochrome c into the cytoplasm. The released cytochrome c binds to the cytochrome c/Apaf-1 complex and oligomerizes with carspace-9 to produce apoptosomes, then activates the carspace cascade containing carspace 3 [51]. Activated caspase-3 cleaves polyADP-ribose polymerase (PARP), causing DNA segmentation, which eventually causes cells to undergo apoptosis [52]. Chrysophanol triggers mitochondrial-mediated apoptosis via the upregulation of the Bax/Bcl-2 ratio and is accompanied by cleavage of caspase-3 and PARP activation in optic nerve meningioma, choriocarcinoma, and breast cancer [33,36,39]. In this study, chrysophanol caused the diminution of Bcl-2 expression, whereas the activations of caspase-3, -7, and -9 and the PARP cleavage were significantly induced in OSCC cells after treatments with chrysophanol. It is suggested that chrysophanol causes apoptosis via caspase activations in OSCC cells (Figure 3).

In the present study, it was confirmed that chrysophanol induces the formation of AVOs and autophagic vacuoles within the cytoplasm or perinuclear regions in OSCC cells. Autophagic biochemical marker expression, such as that of ATG5, beclin-1, LC3, and p62/SQSTM1, has also been verified. The autophagy process is encoded by autophagy-related genes (ATGs), and it leads to the formation of autophagosome [53]. To initiate the autophagosome membrane elongation process, the ATG5-ATG12 complex’s conjugation is necessary [54]. Early studies suggested that beclin-1 (BECN1, also known as ATG6) plays a role in one step of the autophagy process. Beclin-1–Vps 34 complex I promotes the production of autophagosomes [55,56]. The soluble forms of LC3B-I and lipid phosphatidylethanolamine (PE) converted to the autophagic vesicle-associated form, LC3B-II (also known as MAP1LC3B) [57]. For a mature autophagic flux, p62/SQSTM1 (sequestosome-1) targets autophagosomes and links ubiquitinated proteins, and then cellular contents are targeted by sequestosome-1 and degraded in the lysosomes [58]. The upregulated expression level of protein and mRNA, which is associated with autophagy, suggests induced autophagy in OSCC cells when treated with chrysophanol. Chrysophanol-treated cells induced the protein expression levels of beclin-1 and ATG5, and LC3-I was converted to LC3-II. The mRNA expression levels of ATG5, p62/SQSTM1, and MAP1LC3 were considerably induced (Figure 4). This result indicates that chrysophanol significantly induces autophagy in OSCC cells.

The autophagy cell death mechanism remains highly debatable; however, massive autophagy can kill cells. Therefore, autophagy can be considered as a type of cell death [59]. 3-MA was used to further investigate the role of autophagy in chrysophanol-induced apoptosis. 3-MA is popularly used as an autophagy inhibitor. It has been reported to degrade the formation of the pre-autophagosome, autophagosome, and autophagic vacuole by blocking phosphoinositide 3-kinase (class Ⅲ PI3K) activities [60]. In this study, combined chrysophanol and 3-MA treatments showed increased activation of caspase-3 and PARP cleavage and greater release of cytochrome c; on the contrary, the expression levels of beclin-1 and LC3B-II reduced in comparison to a single treatment of chrysophanol. Therefore, autophagy inhibitors increased chrysophanol-induced apoptosis (Figure 6). The factors that regulate apoptosis and autophagy have been revealed. Beclin-1 is a component of the PI3K III complex demanded for autophagosome formation, and it also binds to Bcl-2 or Bcl-XL via the BH3 domain and then disrupts autophagic activity. Beclin-1 is a well-known protein that includes autophagic inducers and autophagic inhibitors [61,62]. In another report, the knockdown of beclin-1 was found to expedite apoptosis [63]. In this study, chrysophanol induced apoptosis and autophagy in oral cancer cells. Among them, autophagy induced by chrysophanol is a cytoprotective mechanism of oral cancer cells and plays a role in partially interfering with the apoptosis pathway. The PI3K/Akt/mTOR pathway plays crucial roles in the regulation of autophagic activity. In the regulation of autophagy, the PI3K/Akt/mTOR pathway has been known as a significant player in catabolic processes. [64,65]. Therefore, in this study, we investigated the phosphorylation of the Akt/mTOR pathway in chrysophanol-treated OSCC cells (Figure 6).

Chrysophanol promotes the phosphorylation of Akt in JEG-3 cells [36], and chrysophanol inhibits the phosphorylation of PI3K/Akt in SNU-C5 cells activated by EGF (epidermal growth factor) [34]. In this study, pan-PI3K inhibitor LY294002, which has been known to inhibit autophagy, was used for verifying chrysophanol-regulated PI3K/Akt pathway activations. PI3K is demanded for autophagic sequestration. Thus, the inhibition of PI3K with LY294002 can hinder autophagic processes [64]. In this study, it was confirmed that the co-treatment of chrysophanol and LY294002 significantly increased pro-apoptotic proteins Bax, cleaved caspase-3, and cleaved PARP while decreasing the conversion of autophagic-related proteins beclin-1 and LC3B-II. That is, the combination of LY294002 with chrysophanol showed the downregulation of autophagy and the upregulation of apoptosis compared with a single treatment of chrysophanol in OSCC (Figure 7).

Taken together, chrysophanol reduced cell viability and induces apoptosis, demonstrating its effectiveness as an anticancer therapeutic, but unfortunately, chrysophanol-induced autophagy was found to interfere with OSCC cell apoptosis (Figure 8). To overcome this, autophagy induced by chrysophanol has been shown to be synergistic by increasing apoptosis when inhibited via the regulation of the Akt/mTOR pathway.

## 5. Conclusions

The purpose of the present study was to investigate the mechanism and role of the autophagy induced by a natural substance, chrysophanol, in oral squamous-cell carcinoma cells. Our study demonstrated that chrysophanol induces autophagy and interferes with apoptosis via the PI3k/Akt/mTOR pathway in oral cancer cells. Therefore, in order to increase the value of chrysophanol as an oral cancer treatment, inhibition of autophagy by chrysophanol is essential, which is expected to have a tremendous synergistic effect on overcoming oral cancer. However, further molecular biological mechanisms and in vivo studies are needed to establish a relationship between autophagy and apoptosis by chrysophanol and provide more solid evidence.

## Figures and Tables

**Figure 1 medicina-59-00042-f001:**
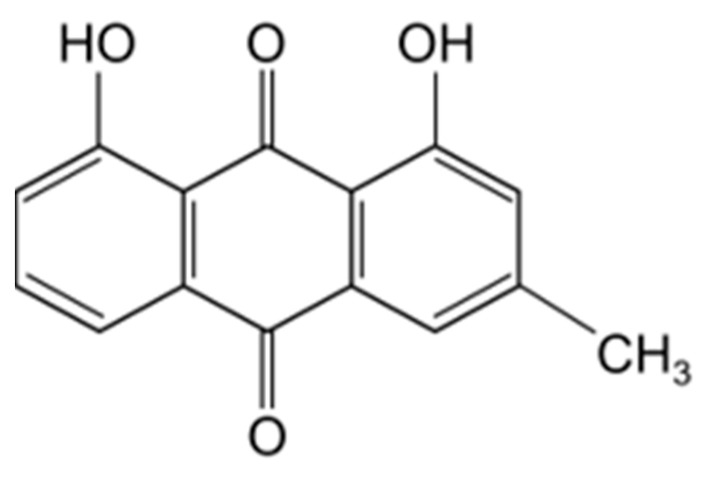
Chemical structure of chrysophanol.

**Figure 2 medicina-59-00042-f002:**
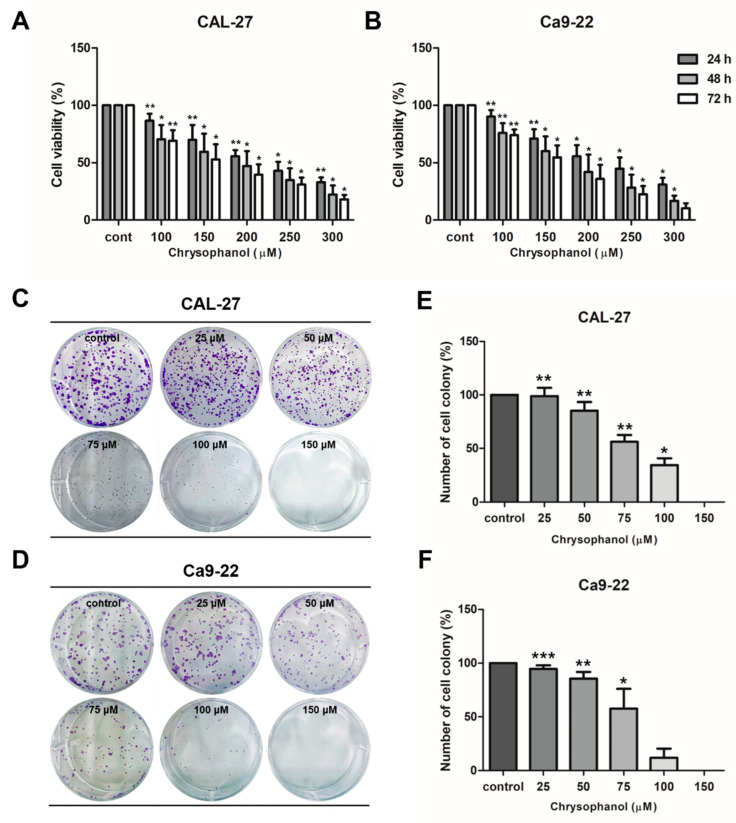
Chrysophanol reduced cell viability and proliferation in CAL-27 and Ca9-22 cells. (**A**,**B**) Both cells were incubated with several concentrations of chrysophanol for 24 to 72 h (0–300 μM), and then the viability of cells was measured using the MTT assay. (**C**–**F**)) A colony-formation assay was performed to examine cell proliferation. Cells were treated with chrysophanol for seven days and stained with a 1% crystal violet solution. The number of colonies was converted into a percentage and shown in a histogram. The results are exhibited as mean ± SD (* *p <* 0.05, ** *p <* 0.01, *** *p <* 0.001).

**Figure 3 medicina-59-00042-f003:**
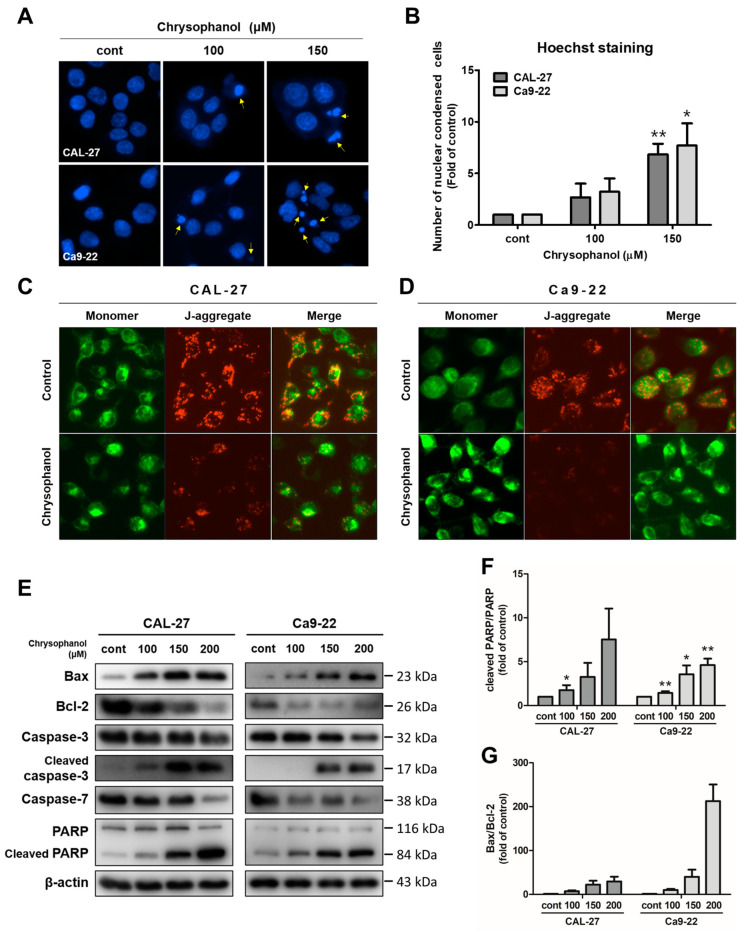
Chrysophanol induced apoptosis via caspase activations in CAL-27 and Ca9-22 cells. Both cells were incubated with chrysophanol (0, 100, and 150 μM) for 24 h. (**A**,**B**). The change in nuclear morphology was investigated by Hoechst agents. Then, nuclear condensation was converted into proportions to control the process and is shown in a histogram. (**C**,**D**) After the treatment of 200 μM chrysophanol, the mitochondrial membrane potential (ΔΨm) was measured by JC-1 solutions. (**E**) Western blot analysis was used to identify the expression level of proteins which affect cell apoptosis, such as Bax, Bcl-2, caspase-3, caspase-7, and PARP. β-actin was used to present a loading control. (**F**,**G**) The relative protein, cleaved PARP/PARP, and Bax/Bcl-2 density were measured; and the fold change was calculated. Data are expressed as mean ± SD (* *p <* 0.05, ** *p <* 0.01).

**Figure 4 medicina-59-00042-f004:**
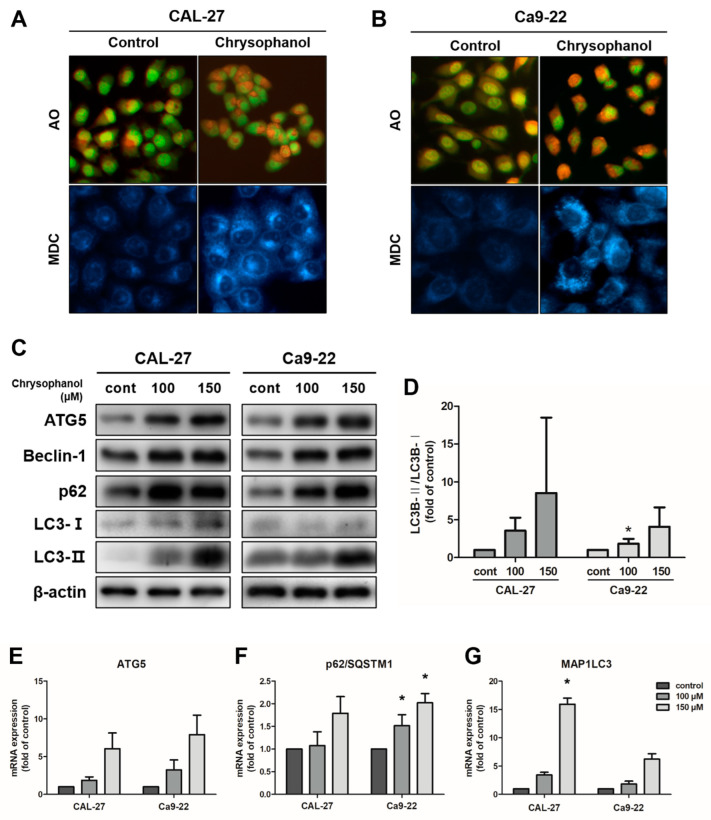
Chrysophanol-induced autophagy in CAL-27 and Ca9-22 cells. (**A**,**B**) The cells were treated with 100 μM chrysophanol. Then, acidic vesicular organelles’ (AVOs) formation was observed by AO staining, and MDC staining was used to detect the autophagic vacuole accumulation. (**C**,**D**) The expression levels of autophagy-related protein ATG5, beclin-1, LC3B-I/LC3B-II, and p62 were examined using a Western blot analysis, and the density of protein LC3B-II/LC3B-I ratios is shown in the graph. β-actin was used to present a loading control. (**E**–**G**) Autophagy-related mRNA expression levels, such as those of ATG5, beclin-1, p62/SQSTM1, and MAP1LC3B, were identified using a real-time PCR. GAPDH was used to present a loading control. Data are expressed as mean ± SD (* *p* < 0.05).

**Figure 5 medicina-59-00042-f005:**
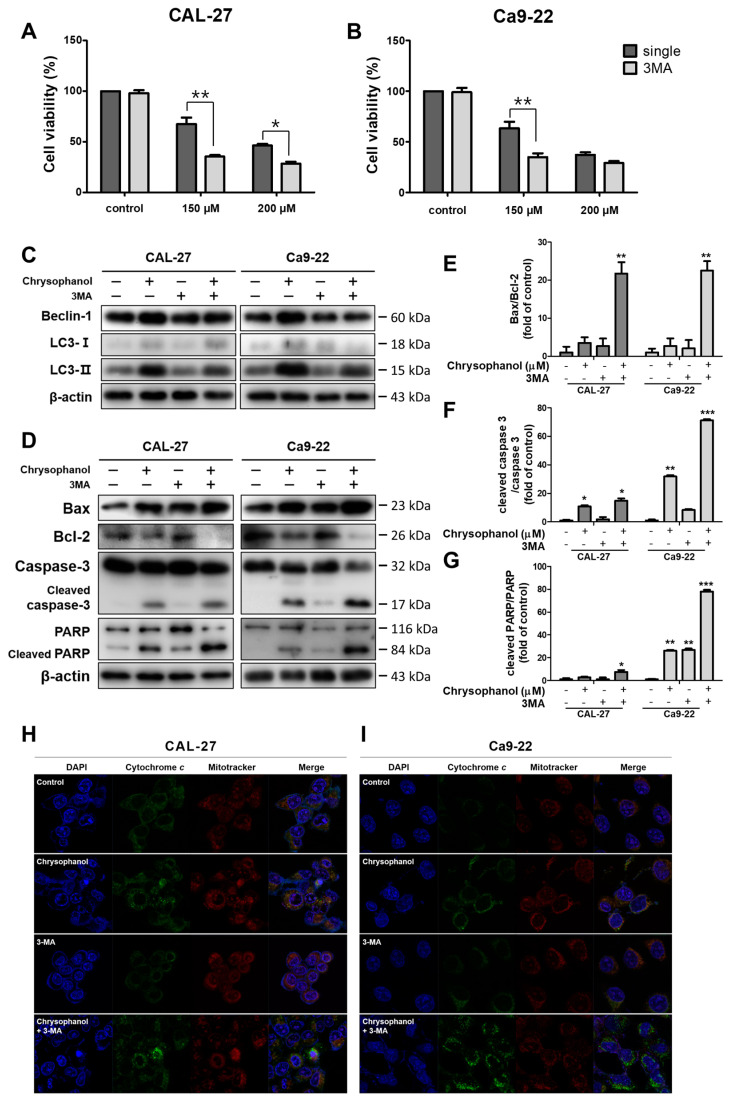
It was found that 3-MA inhibited autophagy and cooperated with apoptosis, which was occasioned by chrysophanol in CAL-27 and Ca9-22 cells. (**A**,**B**) Cell viability was executed by an MTT assay after chrysophanol treatments with or without 3-MAfor 24 h. (**C**,**D**) The protein expression levels of autophagy-associated proteins (ATG5, beclin-1, and LC3B) and apoptosis-associated proteins (Bax, Bcl-2, Caspase -3, and PARP) were detected by using a Western blot analysis. β-actin was used to present a loading control. (**E**–**G**) Bax/Bcl-2, cleaved caspase -3/caspase -3, and cleaved PARP/PARP ratios were calculated by using the Western blot band density. (**H**,**I**) Chrysophanol (0, 150 μM) was treated with both cells in the presence or absence 3-MA, and then the revelation of cytochrome *c* was visualized by immunofluorescence analyses. The results are exhibited as mean ± SD (* *p <* 0.05, ** *p <* 0.01, and *** *p <* 0.001).

**Figure 6 medicina-59-00042-f006:**
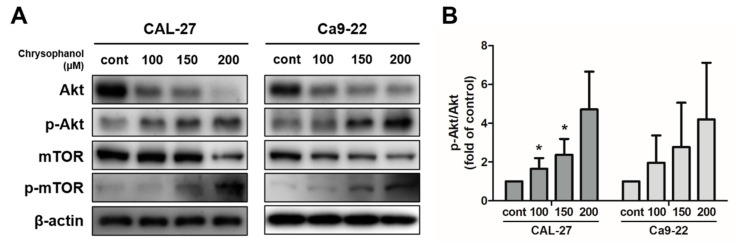
Chrysophanol enhanced the Akt/mTOR pathway in CAL-27 and Ca9-22 cells. (**A**,**B**) Chrysophanol was applied at 0 to 200 μM for 24 h in both OSCC cells. The expression of levels of Akt, p-Akt, mTOR, and p-mTOR were obtained using Western blot analysis and the p-Akt/Akt Western blot band density ratio is shown in the histogram. β-actin was used to present loading controls. Data are expressed as mean ± SD (* *p <* 0.05).

**Figure 7 medicina-59-00042-f007:**
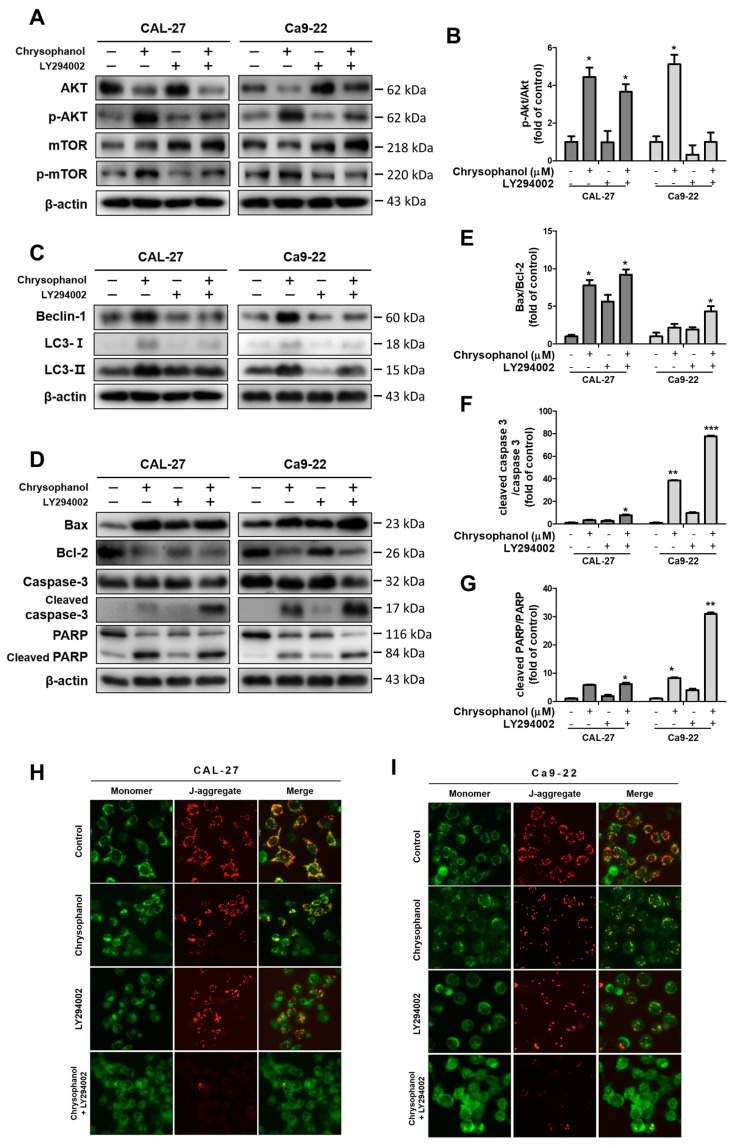
Inhibition of the PI3K/Akt/mTOR pathway promoted chrysophanol-induced apoptosis in CAL-27 and Ca9-22 cells. Both cells were pre-treated with 20 μM LY294002 for 2 h and then treated with chrysophanol for 24 h. (**A**,**B**) The expression levels of Akt, p-Akt, mTOR, and p-mTOR were detected using Western blot analyses, and the relative protein p-Akt/Akt ratio was calculated by a Western blot band. (**C**–**G**) The protein expression levels of autophagy-related proteins (beclin-1, p62, and LC3B) and apoptosis-related proteins (Bax, Bcl-2, caspase -3, and PARP) were evaluated by Western blot analyses, and then the ratios of apoptosis-regulated proteins are indicated in the histogram. (**H**,**I**) Making observations for JC-1, cells were treated with 150 μM chrysophanol in the presence or absence of LY294002. Data are expressed as mean ± SD (* *p <* 0.05, ** *p <* 0.01, *** *p <* 0.001).

**Figure 8 medicina-59-00042-f008:**
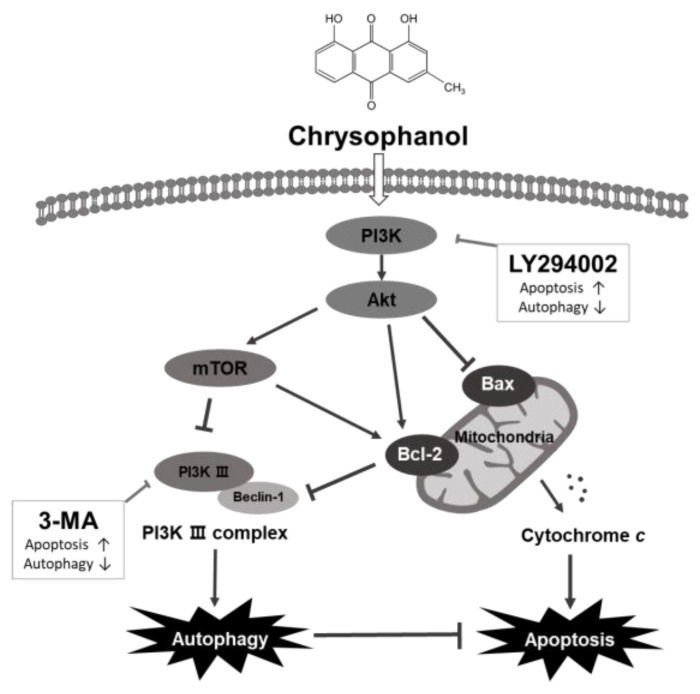
Graphical illustration of apoptotic and autophagic effects induced by chrysophanol in human OSCC, CAL-27 and Ca9-22 cells. Chrysophanol activates autophagy and in parallel inhibits apoptosis activity. 3-MA, an autophagy inhibitor, inhibits autophagy and cooperates with apoptosis, which is occasioned by chrysophanol. Inhibition of the PI3K/Akt/mTOR signaling pathway with the PI3K inhibitor LY294002 enhances chrysophanol-induced apoptosis and reduces autophagy. Therefore, chrysophanol-induced autophagy acts as a cell protection mechanism and interferes with the apoptosis pathway of OSCC cells.

**Table 1 medicina-59-00042-t001:** Sequences of primers.

Target Gene		Primer Sequence (5′ to 3′)
ATG5	Forward	GGGGTGACTGGACTTGTTG
	Reverse	CACTTCCCGCCCTCTGGTATC
p62/SQSTM1	Forward	AGCTCAGGAAGGTGCCATT
	Reverse	TTCTCAAGGCCCCATGTTGCAC
MAP1LC3B	Forward	AAGGCTTTCAGAGAGACCCTG
	Reverse	TTGCGCTTCCAACTCAGGC
GAPDH	Forward	GACAGTCAGCCGCATCTTCT
	Reverse	GCGCCCAATACGACCAAATC

## Data Availability

The data presented in this study are available upon request from the corresponding author.

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
