# Peer review of "Chrysophanol-Induced Autophagy Disrupts Apoptosis via the PI3K/Akt/mTOR Pathway in Oral Squamous Cell Carcinoma Cells"

_medicina, 2022, doi:10.3390/medicina59010042_

Round 1

Reviewer 1 Report

As a summary the manuscript needs to be rewritten since the quality of writing is poor. Authors also need to justify comments provided by the reviewer:

Below are detailed comments from the reviewer:

Statements

Comments

Line 51; explanation on Programmed Cell Death Types/Classification needs to be clearly justified. Sentences written are quite confusing.

Please rewrite statement

Many grammatical errors and poor use of English were found.

Kindly improve the quality of writing.

Line 69 to 72; it is hard to understand what authors are trying to explain. Authors stated “Meanwhile, on a few of occasions, a combined of apoptosis and autophagy are able to be present in reaction to stimuli

[16]”. What kind of reaction stimuli authors refer to? 

Please rewrite statement

Line 98 to 103; “The anti-cancer effect of chrysophanol has revealed to increase apoptosis in several 

cancer cells. By the way, the experiment about chrysophanol effect on apoptosis and autophagy has not been probed in OSCC cells. Moreover, which pathway caused apoptosis and autophagy by chrysophanol was not known in OSCC cells until now. In this research, we were investigated anti-cancer effect of chrysophanol through apoptosis and autophagy on 

CAL-27 and Ca9-22 cells via PI3K/Akt/mTOR signaling pathway. 

In the reviewer point of view, authors would want to explain about problem statements/or gaps in research as well as their research objective. However, the message was not clearly written. Please rewrite.

For statistical analysis, did authors utilize post hoc tests for ANOVA?

Self-reflect for proper statistical analysis 

For figure 4, did all experiments performed at 24 hr? 

Will this short observation be enough to drive the conclusion that Chrysophanol treatment to OSCC cells induced authophagy?

For figure 6; authors stated that “Chrysophanol prompted phosphorylation of Akt dose-dependently in both cells. The 

protein expression levels of p-Akt and p-mTOR were increased dose-dependently in both cells (Figures 6A and 6B).” However, the authors did not state the period for that particular observation performed.

Please state the observation period.

Reviewer concern about the very brief period of time used for observation in relation with experiments shown in figure 7. “Inhibition of PI3K/Akt/mTOR pathway promoted chrysophanol-induced apoptosis in CAL-27 and Ca9-22 cells. Both cells were pre-treated with 20 μM LY294002 for 2 h then treated with chrysophanol.”

Treatment end point for final statistical analysis on cell culture model was not clearly justified.

Figure 7; B, E, F are there any possibility that the significant symbol ** missing?

Please check

Reviewers suggest that you can provide figures to summarize the author's research findings.

Please provide/add figure

Author stated in line 401; 

“chrysophanol induced both apoptosis and autophagy in oral cancer cells, and chrysophanol-induced autophagy was found to play a role in interfering with apoptosis rather than promoting apoptosis.”

Statements seem confusing. Do authors mean that Chrysophanol may allow alternative activation of autophagy-induced cancer cells activation / resistance? Will this then bring to the suggestion that usage of Chrysophanol should be used together with an autophagy inhibitor?

The Conclusion part was poorly written. The message is not clear.

Please rewrite

Author Response

  • We deeply appreciated for your interest in our manuscript and your reviewer’s valuable advices and comments. We are deeply sorry about missing some of the mistakes and submitting the manuscript. The comments were very helpful in improving the manuscript. We fully revised the manuscript as you advised. You will find that we corrected the manuscript according to all suggestions.

Reviewer 2 Report

The manuscript aimed to study the interaction between chrysophanol, apoptosis, and autophagy in OSCC cells. The authors are using an interesting approach, although the effect of chrysophanol in the different cancer cells (including OSCC cells) is well studied. Moreover, it is impossible to reach a conclusion when the experiments were done at different chrysophanol concentrations and with incomplete statistics.

Review comments.

Line 52-53 "Apoptosis arises during a homeostatic reaction to sustain of cell populations and normal development" What is the meaning of "to sustain"?

Line 57-58 "Autophagy is a catabolic function that advanced by a various of cellular stress such as starvation, damage of organelle [11]." What is the meaning of "that advanced"?

The Materials and methods are not described well enough to reproduce the experiments. Also, there is information on the methods used in the results section.

In 2.3 Cell Viability Assay section, what is the wavelength used?

Specify the control used in the experiments. Also, it is desirable to add a positive control to the experiments.

From Figure 2, it is clear that increasing the treatment time from 24 to 48 h, the cell viability is affected, and over 200 µM of chrysophanol concentration 50% of viability is reached. Then why were some experiments conducted at 18 and 24 h of treatment with chrysophanol?  Why was a chrysophanol concentration below 200 µM used? Moreover, why was the experiment conducted at different chrysophanol concentrations? What was the LC50 obtained?

Add the treatment time and chrysophanol concentration used in every experiment.

The statistics are incomplete. Add the standard deviation and significance to all data.

Line 280-282 "Cells which single treatment of chrysophanol was shown a higher rate of cell viability than combination of chrysophanol and 3-MA (Figures 5A and 5B)." Although, Figure 5 shows that the addition of 3-MA has no statistical significance in cell viability, even between 150 and 200 µM of chrysophanol concentration (Figure 5A). 

The axes' names are unclear in Figures 5E-5G, 7B, 7E-7G.

Figures 5H-5I are not discussed in the manuscript.

It is unclear why the selected proteins and genes were used in the Western Blot Analysis and the Gene Expression Analysis. Add the information in the Material and Methods section. Adding a diagram or model of the signaling pathway with the effects of chrysophanol is recommended.

Author Response

(The authors gave the same response as above.)

Reviewer 3 Report

Journal of medicina

Research Article;

The article entitled “Chrysophanol-induced Autophagy Disrupts Apoptosis via the PI3K/Akt/mTOR Pathway in Oral Squamous Cell Carcinoma Cells’’. The author investigate the Natural products which is a necessary source for drug discovery and contribute to cancer chemotherapy over the past decades. Chrysophanol have anticancer effect of in several cancer cells, it is good to study the effect of Chrysophanol in oral squamous cell carcinoma cells. The study found that chrysophanol impedes the growth and viability of CAL-27 and Ca9-22, and induces apoptosis through the intrinsic pathway. They also invistigate that chrysophanol activates autophagy-related factors (ATG5, Becelin-1, P62/SQSTM1) and LC3B conversion. chrysophanol also activated both apoptosis and autophagy. chrysophanol confirmed that the rate of apoptosis was significantly increased. According to the study the potential of chrysophanol as an excellent anti-cancer agent in Oral Squamous Cell Carcinoma was confirmed through this study.

I carefully read the manuscript and found it suitable for publication in the journal. I accept this article for possible publication. There are some common mistakes in the article which should be corrected by the authors. After the correction of all the mistakes, the article could be considered for publication in the prestigious medicina Journal.

Comments for Authors

Ø  Write keywords in alphabetical order.

Ø  Section Introduction; The author’s needs to put more latest related citations in the introduction part.

Ø  2. Materials and Methods (2.3. Cell Viability Assay) the author need to revise it with cell proliferation assay.

Ø  It will be batter to check the effect of Chrysophanol by flow cytometry by using Annexin-V.

Ø  The author needs to revise the Hoechst staining of figure-3A. And make it more clear the nuclear accumulation.

Ø  Mentioned the size of the image taken during all figure.

Ø  Revised all the Figure significance value and mentioned in legend (***p<0.001, **p<0.01,*p<0.05). I-e in Figure-2, the represent *** but in legend present only **.

Ø  Mentioned the original dimension clearly all Figures.

Ø  Write the absorbance M.W of protein in each western blot result image.

Ø  It will be batter to represent in graphical image of the manuscript.

Ø  Discussion section. The first paragraph of (line number 332-342) is useless. Although in introduction apoptosis and autophagy is described and the author needs to revised.

Ø  Use EndNote or Mendeley software for references sequences.

Ø  Check grammatically and spelling throughout the manuscript. There are some mistakes.

Cite the following references;

v  DOI: 10.2174/1871520622666220831124321

v  DOI: 10.1038/s41419-021-03771-z

v  DOI: 10.2174/1386207324666210216094428

v  DOI: 10.1016/j.bioadv.2022.213039

Author Response

(The authors gave the same response as above.)

Round 2

Reviewer 1 Report

Dear authors,

Reviewers feel that few corrections need to be made although authors have covered almost all aspects that were queried.

Few points that need corrections/amendments are:

  1. The hypothetical figure should include a brief explanation of the mechanism induced by chrysophanol.

  2. The conclusion seems not clear. Please rewrite. From our understanding, chrysophanol has activated autophagy and in parallel inhibits apoptosis activity. This will inhibit cancer cells from undergoing extensive cell death and will be an alternative pathway activation for cancer cell survival. Do authors think chrysophanol has an advantage?

It hope that authors can justify these matters soonest.

Thank you.

Author Response

Dear authors,

Reviewers feel that few corrections need to be made although authors have covered almost all aspects that were queried.

Few points that need corrections/amendments are:

  1. The hypothetical figure should include a brief explanation of the mechanism induced by chrysophanol.

Thank you for your valuable comments. According to your comment, the figure of Fig8 has been partially modified and the figure legend has been added as follows.

Figure 8. Graphical illustration of apoptotic and autophagic effects induced by chrysophanol in human OSCC, CAL-27 and Ca9-22 cells. Chrysophanol activates autophagy and in parallel inhibits apoptosis activity. 3-MA, an autophagy inhibitor inhibits autophagy and cooperates with apoptosis, which is occasioned by chrysophanol. Inhibition of the PI3K/Akt/mTOR signaling pathway with the PI3K inhibitor LY294002 enhances chrysophanol-induced apoptosis and reduces autophagy. Therefore, chrysophanol-induced autophagy acts as a cell survival mechanism and interferes with the apoptosis pathway of OSCC cells.

  1. The conclusion seems not clear. Please rewrite. From our understanding, chrysophanol has activated autophagy and in parallel inhibits apoptosis activity. This will inhibit cancer cells from undergoing extensive cell death and will be an alternative pathway activation for cancer cell survival. Do authors think chrysophanol has an advantage?

→ Thank you for your valuable opinion. As your commnet about the conclusion section, we identified the controversial part and revised the sentence as follows.

The purpose of present study is to investigate the mechanism and role of the autophagy induced by the natural substance, chrysophanol in oral squamous cell carcinoma cells. Our study demonstrated that chrysophanol induces autophagy and interferes apoptosis via PI3k/Akt/mTOR pathway in oral cancer cells. Therefore, in order to increase the value of chrysophanol as an oral cancer treatment, inhibition of autophagy by chrysophanol is essential, which is expected to have a tremendous synergy effect in overcoming oral cancer. However, further molecular biological mechanisms and in vivo studies are needed to establish a relationship between autophagy and apoptosis by chrysophanol and provide more solid evidence.

It hope that authors can justify these matters soonest.

Thank you.

We sincerely thank you for enhancing the clear academic value of our manuscript by your review.
